# Magnetic susceptibility properties of tumor-associated cells imaged by MRI reveal glioblastoma infiltration in the edema region

Giulia Debiasi[1,2,3], Giovanni Librizzi[4,5], Valentina Visani [2], Marco Castellaro[2], Zhenghao Li[6], Hongjiang Wei[6], Renzo Manara[4,5,7], Alessandra Bertoldo[2,4] & Chunlei Liu [1,8] ✉

## Abstract

**Background** Glioblastoma is a malignant primary brain tumor. Because of its highly invasive and infiltrative nature, surgical resection and radiation therapy are not able to remove all tumor cells, even with state-of-the-art imaging and fluorescence-guided surgery.
**Methods** 24 newly diagnosed glioblastoma patients were enrolled. Pre- and post-surgery MRI scans were performed. Magnetic susceptibility was quantified based on gradient echo MRI. The ratio between sub-voxel paramagnetic and diamagnetic susceptibility components was computed. Relationships between the proposed ratio metric and prognostic factors and pathological iron were investigated. Perfusion and permeability imaging were used to exclude the presence of blood-related contribution to the paramagnetic component.
**Results** Here we show that by decomposing tissue magnetic susceptibility into paramagnetic and diamagnetic sources, we can identify, non-invasively and in vivo, areas of altered iron metabolism associated with tumor activities in the edema tissue surrounding glioblastoma. We find that the paramagnetic to diamagnetic susceptibility ratio uniquely delineates area of hyperintensity corresponding to a Tumor and Immune cells Infiltration Zone. Statistically significant relationships are found between the ratio metrics in the infiltration zone and tumor prognostic factors. Follow-up scans reveal tumor progression and later contrast-enhancement in the predicted infiltration zone. Histological data indicate that increased iron content causes the elevated ratio metric.
**Conclusions** Our study proposes a method to derive an iron-related imaging marker of abnormal patterns in the edema region of the glioblastoma associated with tumor cell infiltration. We show the potential of the imaging marker to aid and improve surgical and treatment planning.

## Plain language summary

Glioblastoma is a brain tumor with very infiltrative behavior. Current treatment options are not able to remove it in its entirety. Consequently, residual tumor is found in most patients after surgery, causing early recurrence and decreased survival. We focused on the edema region of the tumor, usually homogeneous on conventional MRI, to identify potential areas of infiltration. By computing the tissue magnetic properties using MRI data, we observed abnormal spatial patterns in edema. We developed a method to automatically identify these abnormal areas. We found an association between the magnetic properties of those areas and tumor prognostic factors, which is caused by an elevated iron presence in the tumor tissues. We further showed that this method may be used to track tumor infiltration. Our method could be readily employed in clinical practice to aid surgical resection and treatment planning.

Glioblastoma (IDH-wild-type) is the most malignant type of primary brain tumor. Despite multimodal treatment approaches including surgery, radiotherapy and chemotherapy, the median overall survival is about 14.6 months[1,2]. Magnetic Resonance Imaging (MRI) is routinely used in clinical practice at various stages of the management of glioblastoma, from initial diagnosis to treatment planning and monitoring. In most cases, the contrast-enhancing region identified with MRI is used to guide surgery and to provide a reference for radiotherapy planning. Incomplete resection of the tumor is associated with shorter survival of the patients[3,4], and a recent report has highlighted the potential patient survival benefits derived from

[1]Department of Electrical Engineering and Computer Sciences, University of California, Berkeley, CA, USA. [2]Department of Information Engineering, University of Padova, Padova, Italy. [3]Department of Surgery, Oncology and Gastroenterology, University of Padova, Padova, Italy. [4]Padova Neuroscience Center, University of Padova, Padova, Italy. [5]Neuroradiology, Department of Neurosciences, University of Padova, Padova, Italy. [6]School of Biomedical Engineering, Shanghai Jiao Tong University, Shanghai, China. [7]Department of Medicine, University of Padova, Padova, Italy. [8]Helen Wills Neuroscience Institute, University of California, Berkeley, CA, USA. ✉e-mail: chunlei.liu@berkeley.edu

resection of non-enhancing tissue during surgery[5]. While edema and non-enhancing regions of the tumor are potential sites of tumor infiltration[4,6,7], usually they are not included in surgical resection, as routine MRI cannot differentiate tumorous tissues in those regions. Intraoperative fluorescence imaging with 5-aminolevulinic acid (5-ALA) can identify tumoral cells and macrophages[8] beyond the contrast-enhancing tissue identified with MRI[9,10]. However, its utility is limited by optical visibility.

Recent studies have investigated the heterogeneity of edema and non-enhancing tissue in glioblastoma with different MRI modalities. One study suggested that heterogeneity in peritumoral edema exists between recurrence and non-recurrence subregions based on radiomic features derived from clinical MRI sequences, which was trained on 22 patients and verified on two individuals[11]. Similarly, it was suggested that clustering radiomic features combined with $O^6$-methylguanine-DNA methyl-transferase (MGMT) methylation status may improve survival prediction[12]. In addition, diffusion MRI techniques such as Neurite Orientation Dispersion and Density Imaging (NODDI)[13] and Diffusion Tensor Imaging (DTI) have been applied to differentiate between edema and non-enhancing tissues[14]. Fiber density and structural brain connectome were further suggested to provide information on glioblastoma cells infiltration[15]. In addition to MRI, 18-kDa mitochondrial translocator protein (TSPO) Positron Emission Tomography (PET) imaging has also been reported to be a useful marker of tumor and immune cells in glioblastoma, even outside the contrast-enhancing region of the tumor on MRI[16]. Nevertheless, these MRI-based techniques have mostly relied on group analysis and so far, lack the sensitivity and specificity needed for clinical utility to identify tumor infiltration outside the contrast-enhancing region of the glioblastoma.

Tumor cell proliferation is strongly dependent on iron, resulting in an altered iron metabolism. It is a complex process that involves multiple stages, such as iron uptake, storage, utilization, and release. Both tumor and immune cells are rich in iron content. Tumor-associated macrophages (TAMs) can store and release iron depending on their subtypes[17–20]. Cellular iron is generally paramagnetic, exhibiting positive magnetic susceptibility, a physical quantity that describes the change in magnetization of a material in response to an externally applied magnetic field[21]. Magnetic susceptibility can be quantified with MRI using the recently developed Quantitative Susceptibility Mapping (QSM) technique[21,22], which exploits frequency shifts measured with gradient echo pulse sequence. It is hypothesized that mapping iron in lesioned regions may provide an indication of the presence of tumor activities. In gliomas, QSM has already shown that magnetic susceptibility in the enhancing and necrotic tissues is correlated with both ferritin light chain and macrophage counts[23]. Also, identification of oligodendrogliomas among IDH-mutant gliomas was achieved with high specificity based on magnetic susceptibility[24]. Relatedly, QSM has been suggested as a potential imaging marker for the assessment of IDH genotypes, in combination with the Apparent Diffusion Coefficient (ADC) measured by diffusion MRI[25].

Besides paramagnetic compounds such as iron, cells and tissues also contain diamagnetic compounds such as myelin, calcification and lipids, of which excessive production and accumulation have been reported in many cancers. However, QSM is only able to measure the average magnetic susceptibility in a voxel of an MRI image where the contributions of paramagnetic and diamagnetic materials can cancel each other out. To overcome the issue, among others, the method DECOMPOSE-QSM[26] computes a Paramagnetic Component Susceptibility (PCS) and a Diamagnetic Component Susceptibility (DCS), which respectively provide information related to iron and the other diamagnetic compounds within a voxel.

We hypothesize that quantifying the relative contributions of paramagnetic and diamagnetic susceptibility could provide new insights into the heterogeneity of the regions outside the contrast-enhancing tissue (i.e., edema and non-enhancing regions) of glioblastoma. We have identified a potential MRI-based imaging marker whose underpinnings are related to the altered iron metabolism of tumors and that could be beneficial at different stages of the management of glioblastoma in clinical practice.

## Methods

This research is compliant with the ethical standards of the institutional research committee and with the 1964 Helsinki declaration plus later amendments, and has been approved by the University Hospital of Padova Institutional Review Board. For this study, 24 newly diagnosed glioblastoma, IDH-wild-type patients (WHO 2021) were enrolled at the University Hospital of Padova (Padova, Italy) between March 2023 and June 2024. All subjects provided informed, written consent in accordance with the approved study protocol (protocol number: AOP2971).

If not otherwise specified, processing and analysis were performed in MATLAB R2022b (Mathworks, Natick, MA, USA).

### MRI acquisition

Data were collected on a 3 T Philips Ingenia scanner equipped with a 32-channel head-neck coil. For susceptibility quantification, a 3D multi-echo GRE sequence (eight echoes $TE_1/\Delta TE = 5/5$ ms, TR = 44 ms, FA = 25°, voxel size = $1 \times 1 \times 1$ mm³, FOV = $240 \times 240$ mm², 140 slices) was acquired. The acquisition protocol also comprises conventional structural MRI sequences: (i) 3D T1-weighted (T1w) image pre- and post-gadolinium-based contrast agent injection (T1w-Gd) (TE = 3 ms, TR = 6.7 ms, FA = 8°, voxel size = $1 \times 1 \times 1$ mm³, FOV = $240 \times 240$ mm², 181 slices); (ii) 3D T2-weighted (T2w) image (TE = 280 ms, TR = 3000 ms, FA = 90°, voxel size = $1 \times 0.94 \times 0.94$ mm³, FOV = $256 \times 256$ mm², 181 slices); (iii) 3D Fluid Attenuated Inversion Recovery (FLAIR) image (TE = 360 ms, TR = 8000 ms, FA = 90°, voxel size = $1.12 \times 1.12 \times 1.12$ mm³, FOV = $221 \times 221$ mm² - reconstructed to voxel size = $0.56 \times 0.62 \times 0.62$ mm³, FOV = $400 \times 400$ mm², 326 slices). In addition, multi-shell diffusion MRI protocol for a total of 116 Diffusion Weighted Images (DWIs) (TE = 104 ms, TR = 3.7 s, FA = 90°, voxel size = $2 \times 2 \times 2$ mm³, FOV = $112 \times 122$ mm², 78 slices, multi-band accelerator factor = 2) was acquired: 12 images at b = 0 s/mm², 8 gradient directions at b-value = 300 s/mm², 32 gradient directions at b-value = 1000 s/mm² and 64 gradient directions at b-value = 2000 s/mm². Moreover, to quantify blood perfusion and permeability, both Dynamic Susceptibility Contrast (DSC) and Dynamic Contrast Enhanced (DCE) sequences were performed. DCE sequence involved a variable flip angle technique (VFA) and 80 dynamic scans with the same geometry (TE = 1.8 ms, TR = 5 s, voxel size = $2.4 \times 2.4 \times 5$ mm³, FOV = $96 \times 96$ mm², 22 slices). Images for T1 mapping using VFA were acquired (FA = 5°/10°/15°), and after intravenous administration of one bolus of gadolinium-based contrast agent (dosage: 3 ml, injection rate: 2 ml/s), dynamic DCE images were collected (FA = 15°, 80 volumes). In between DCE and DSC acquisitions, there was a time interval of 30 s. Then, DSC acquisition started and after 45 s a second bolus (dosage: 7 ml, injection rate: 5 ml/s) was administered (TE = 40 ms, TR = 1.5 s, FA = 75°, voxel size = $2.4 \times 2.4 \times 5$ mm³, FOV = $94 \times 94$ mm², 22 slices, 120 volumes).

### Structural images pre-processing

T1w, T2w, FLAIR and T1w-Gd were bias field corrected[27], brain extracted[28] and linearly registered to the native T1w space using the Advanced Normalization Tools (ANTs)[29]. Gray matter (GM) and white matter (WM) segmentations on the T1w image were obtained with SPM12[30], by setting thresholds for the probability maps of GM and WM to 0.7 and 0.7, respectively. To avoid contamination by the tumors, only the GM and WM segmentation belonging to the contralateral hemisphere were considered. Diffeomorphic non-linear registration of the T1w image to the symmetric MNI152 atlas was achieved with ANTs by excluding the lesion area during the estimation of the transformation[31]. Subcortical segmentation on the T1w image was obtained with FSL. The T1w image was linearly registered to the first echo magnitude image with ANTs. The estimated transformation was applied to GM, WM and subcortical structures masks. Segmentation and mask were visually inspected and manually edited if a clear mismatch was observed.

## Tumor segmentation

Tumor masks were generated with an automatic tool[32,33] based on T1w, T2w, FLAIR and T1w-Gd, which classified the tumor area into three different tissue types, namely (1) edema/non-enhancing tumor (ED); (2) necrotic core (NCR); (3) contrast-enhancing tumor (ET). Masks were then visually inspected in the native T1w space and modified (if necessary) by neuroradiologists with experience in the field of gliomas.

The corresponding masks in the contralateral area were derived by exploiting the symmetric MNI152 atlas. The transformation estimated in the non-linear registration was applied to the tumor mask that was then flipped in the symmetric space and brought back into the native T1w space with the inverse of the same transformation.

The tumor and contralateral masks were also warped to the susceptibility space with the previously estimated transformation for later steps.

## Magnetic susceptibility imaging

Brain extraction was performed with FSL on the first echo magnitude image. QSM maps were obtained for each echo using STISuite (https://chunleiliulab.github.io/software/#STI) as follows. Laplacian phase unwrapping[22] was performed, and then the brain mask was eroded by 3 mm to account for signal drops at the boundaries of the brain. The binarized tumor mask was added back to the eroded brain mask, given the potential erosion of tumor voxels when the tumor is near the cortex. Background field removal was obtained with V-SHARP, and the radius of the spherical mean value filter was set to 12 mm[34,35]. Dipole inversion step was executed with STAR-QSM[36].

DECOMPOSE-QSM algorithm was used to compute PCS and DCS maps. Details about the algorithm can be found elsewhere[26] but, briefly, DECOMPOSE models each voxel signal as the sum of three susceptibility sources, namely paramagnetic, diamagnetic and neutral. A non-linear optimization problem is then solved using the QSM maps and the multi-echo GRE data.

## Perfusion imaging

DSC images were used to compute Cerebral Blood Volume (CBV) maps. A Matlab-based tool (https://github.com/FAIR-Unipd/dsc-mri-toolbox) was applied to obtain CBV values for each voxel. Specifically, the acquired DSC signal was converted to concentration, and the Arterial Input Function (AIF) was extracted with a semi-automatic method[37]. CBV was calculated as the ratio between the Area Under the Curve (AUC) of the tracer concentration and the AUC of the AIF. Then, CBV was corrected for contrast-agent leakage[38]. Baseline DSC volumes were averaged and linearly registered to the T1w image. The inverse of the estimated transformation was applied to the WM mask to normalize CBV maps to the contralateral normal appearing white matter, resulting in nCBV maps. The same transformation was also applied to the tumor and contralateral masks.

## Permeability imaging

DCE images were motion corrected with *mcflirt* in FSL, and image volumes of the VFA sequence were registered to the dynamic images space with rigid transformation. Quantification of permeability parameters was carried out with ROCKETSHIP[39]. In particular, T1 maps were derived from the VFA sequence by a linear fitting, and the Vascular Input Function (VIF) voxels were automatically determined from a predefined area with an automatic algorithm that looks for the typical DCE signal, characterized by a fast rise and a slow decay. The manually delineated area for the VIF was taken from the Superior Sagittal Sinus (SSS) as it has proven to be the best choice for vessel selection in brain tumors[40,41] and given the temporal resolution of the data, the timing difference between arterial and venous blood is negligible[42]. Raw data of the VIF were fitted to a linear upslope and a bi-exponential decay model. Among other parameters, the rate of permeability, $K^{trans}$, was estimated with the extended Tofts model[43] to trade off between model complexity and assumptions, and the temporal resolution of the data. DCE images were linearly registered to the T1w image, and the inverse of the computed transformation was applied to the tumor mask.

## Exclusion of tumor voxels dominated by blood

Perfusion and permeability imaging were used to exclude voxels dominated by blood contribution that are not relevant for the purpose of the present study. Both imaging techniques were used as they provide complementary information about blood.

Perfusion quantification assumes that the contrast agent is intravascular, meaning that the Blood-Brain Barrier (BBB) is intact. Thus, in native DSC space, the area contralateral to the tumor mask was used as reference to determine a threshold for physiological perfusion. For each subject, the 95th percentile of the distribution of nCBV of the contralateral healthy tissues was set as threshold above which corresponding voxels of the tumor mask were considered hyperperfused and therefore excluded from the analysis by creating a binary mask.

Permeability quantification, on the other hand, is only valid when the BBB is disrupted, and the contrast agent goes in the Extracellular Extravascular Space (EES). The T1w-Gd is the imaging modality in which BBB damage is detected, and consequently, the contrast-enhancing tissue is the primary region of interest for permeability assessment. We used DCE acquisition to exclude blood-related voxels due to potential BBB disruption. The contrast-enhancement area was chosen to compute the threshold in the native DCE space. For each patient, the 5th percentile of the distribution of $K^{trans}$ in the contrast-enhancing tissue was calculated, and voxels for which $K^{trans}$ exceeded that threshold were included in the binary mask for exclusion.

The binary mask was warped into the susceptibility space by concatenating transformation among the native space of the masks, the T1w space and the first echo magnitude space. Then, the mask was applied to susceptibility images to exclude voxels dominated by blood.

## PCS vs. DCS ratio (PDR) computation and tumor and immune cells infiltration zone ("TIZ") delineation

The PDR was computed as the ratio between PCS and the absolute value of DCS. To delineate what we call "TIZ," which corresponds to the hyper-intense area on the PCS in edema, WM and GM were isolated in the hemisphere contralateral to the tumor, and subcortical structures were deleted from GM mask, if present. To account for outliers and registration errors, PDR values exceeding the 95th percentile of the distribution of WM and GM were eliminated. Because the GM PDR is larger than the WM PDR, the maximum PDR of the remaining GM PDR distributions was used as the threshold to determine the "TIZ". Specifically, for each voxel in the edema, if its PDR was above the threshold, it was assigned to the "TIZ" region; otherwise, it was not assigned to the "TIZ." As QSM is known to vary significantly between different algorithms, we evaluated the overlap of the resulting "TIZ" derived from STAR-QSM and iLSQR[44] methods by computing the Dice-Sørensen coefficient.

## Diffusion MR imaging

Diffusion volumes affected by interslice instabilities[45] were discarded after visual inspection. Pre-processing was performed with MRtrix software[46] and included denoising and correction for B0 inhomogeneities, eddy currents and motion[47,48]. Registration between the average b0 and the T2w was estimated with ANTs. The ADC was computed as an additional conventional MRI modality used in clinical practice.

## Biological relevance

Relationships between the PDR of the segmented "TIZ" and biological features specific to each patient were investigated as described later in the "Statistical analysis" section. In particular, age was included as a prognostic factor, as well as the MGMT methylation status and the percentage of methylation. Methylation measurements were obtained by Pyrosequencing (PyroMark Q96 ID system, QIAGEN).

## Statistical analysis

For each patient, Wilcoxon rank sum test was carried out between the PDR values of the "TIZ" and the contralateral healthy tissues. The test was

**Table 1 | Patients' demographics and clinical information**

| Sex (M/F) | 18/6 |
|---|---|
| Age at diagnosis (years) | 66 ± 8 |
| MGMT methylation | |
| Subjects, *n* | 19 |
| Methylation status (Y/N) | 13/6 |
| Methylation % | 24 ± 16 |

repeated between PDR in the "TIZ" and the area complementary to the "TIZ" within edema.

A logarithmic transformation was applied to PDR values (logPDR), given their specific distribution, which covers several orders of magnitude. Due to sample size limitations, no subgroup analysis was performed, and we defined only a subset of histogram features to be used, namely median, mode and the 10th percentile of the distributions. Median and mode were chosen to describe the non-normally distributed data, while the choice of the 10th percentile was driven by the distribution of the PDR values, entailing the vast majority of the data on the left side. Association between logPDR-derived features in the "TIZ" and MGMT methylation status was explored with Wilcoxon rank sum test. For age and the MGMT methylation percentage, Pearson's correlation analysis with the logPDR-derived features was performed. In addition, correlations between age and the same histogram features derived from the PCS in the "TIZ" were analyzed to investigate the well-known relationship of susceptibility with age[49]. Also, the correlation between the volume of the "TIZ" and the volume of edema was assessed. If not otherwise specified, the significance level was set to 0.05 for all the above-mentioned analyses.

### Brain tumor patients in vivo MRI, ex vivo MRI and histochemical analysis

Two patients were enrolled specifically to analyze the relationship between iron deposition and elevated PDR values. All the following procedures were approved by the Ethics Committee of Ruijin Hospital Luwan Branch, Shanghai Jiao Tong University School of Medicine. Written informed consent was obtained from the patients. In vivo MRI was performed before surgical resection of the tumor mass, after which a tumor specimen was imaged with a 9.4 T MRI scanner and then processed for histochemical analysis. Specifically, DAB-enhanced Perl's iron staining was performed. A 10-year-old male patient diagnosed with a left temporal lobe neuroepithelial tumor and a 22-year-old male patient diagnosed with a left occipital lobe ganglioglioma (WHO Grade I) were enrolled. Additional details about the MRI acquisition sequences, tumor specimen preparation for imaging and histochemical analysis are provided in the "Supplementary Methods" section.

### Reporting summary

Further information on research design is available in the Nature Portfolio Reporting Summary linked to this article.

## Results

Demographical and clinical information of the 24 newly diagnosed glioblastoma patients enrolled in the present study are summarized in Table 1.

### PCS hyperintensity in edema not seen on conventional MRI

Since we were interested in the iron compound due to its altered metabolism in the tumor, we first focused on the PCS maps. Figure 1 shows MRI and derived quantitative maps from three representative patients. Structural images are used for lesion delineation (colored masks in Fig. 1i), while perfusion and permeability imaging provide blood-related information. The ADC is commonly used in clinical practice as a proxy of cellularity, but it is not specific for biological processes involving tumor and immune cells and visually it does not exhibit clear and consistent delineation of edema regions

of potential high cellularity (Fig. 1e). In contrast, within the edema, regions of hyperintense PCS exhibit a striking contrast over the rest of the tissue which has low PCS values (Fig. 1h PCS only and Fig. 1i, white arrow in the green area on the PCS). This contrast is not visible on conventional MRI (Fig. 1a–e), where the edema appears generally homogeneous. This phenomenon is observed in all patients included in this study, and a representative axial slice of PCS map for each of them is reported in Supplementary Fig. 1.

As QSM reconstruction is prone to streaking artifacts, to ensure that what we observed in the PCS was not the result of artifacts, we evaluated the results in several ways. First, we repeated the QSM processing by changing the dipole inversion algorithm, namely by using the iLSQR algorithm[44] as iLSQR is known to generate somewhat different streaking artifacts than STAR-QSM. In Supplementary Fig. 2a, b, e, f, an example of PCS maps calculated with the two methods is reported. It is seen that the PCS hyperintensity is visible on the PCS maps obtained with both processing methods. We further tested other QSM algorithms on a few randomly selected patients, and the PCS hyperintensity is preserved in all resulting PCS maps. Examples can be found in Supplementary Fig. 2c, d, g, h. Second, we examined the PCS hyperintensity longitudinally. Supplementary Fig. 3 shows a representative subject for which PCS was computed also for the acquisition at three months after surgery. The area of PCS hyperintensity is still observable in the edema corresponding to the homogeneous hyperintense region on the FLAIR image. Third, to confirm that the observed PCS hyperintensity is not caused by potential artifacts resulting from the DECOMPOSE algorithm, we examined the positive susceptibility maps by simply thresholding the QSM maps and compared it against PCS for each subject. A similar contrast is observed in all subjects within the edema. Supplementary Fig. 4 shows an example of axial slice from one subject. A pattern similar to the one found in the PCS can still be noted. Together, these findings support that the observed hyperintense PCS zone within the edema is an intrinsic imaging signature of the tissue rather than potential artifacts.

### Hyperintense PCS in edema is not caused by blood

Hyperintense PCS suggests a high concentration of paramagnetic compounds in those regions. Blood products, due to leaky blood vessels or increased vascularity, are known to introduce high paramagnetic susceptibility. In fact, iron is the main source of paramagnetic susceptibility in the brain[50], but it is known that it can be found as non-heme- or heme-iron[21]. Hemoglobin is a heme-iron complex present in red blood cells, and it influences the magnetic property of blood. To date, it is not possible to distinguish between different sources of iron with susceptibility imaging. To test whether the observed hyperintense PCS is caused by blood vessels, perfusion and permeability imaging were used to quantify nCBV and $K^{trans}$, displayed for each representative patient in Fig. 1f, g, respectively. In particular, $K^{trans}$ is reported in color scale, and it is overlaid on the T1w-Gd to provide anatomical references, as it is quantified only for those voxels where the BBB is disrupted. In all patients, high perfusion and permeability values are mainly observed in the enhancing tissue and its immediate surroundings, but not in the edema. To quantitatively assess the spatial contribution of blood products, nCBV and $K^{trans}$ maps were used to create binary masks of voxels in which blood was identified as the main source of paramagnetic contribution. The masks were used to eliminate voxels that may be contaminated by blood products. Supplementary Fig. 5 reports in boxplots the percentages of tumor voxels that are not excluded during the process of deletion of blood-dominated voxels. Specifically, median and inter-quartile range are 93 and 6 (%) for edema, 49 and 56 (%) for necrosis, and 17 and 16 (%) for the enhancing tissue. Therefore, the vast majority of voxels within the edema is not contaminated by blood, which suggests that the hyperintense PCS area is caused by other cellular sources. This also suggests that even without DSC and DCE acquisition or any other imaging modality related to blood, it would be feasible to identify hyperintense PCS of non-blood origin in the non-enhancing area.

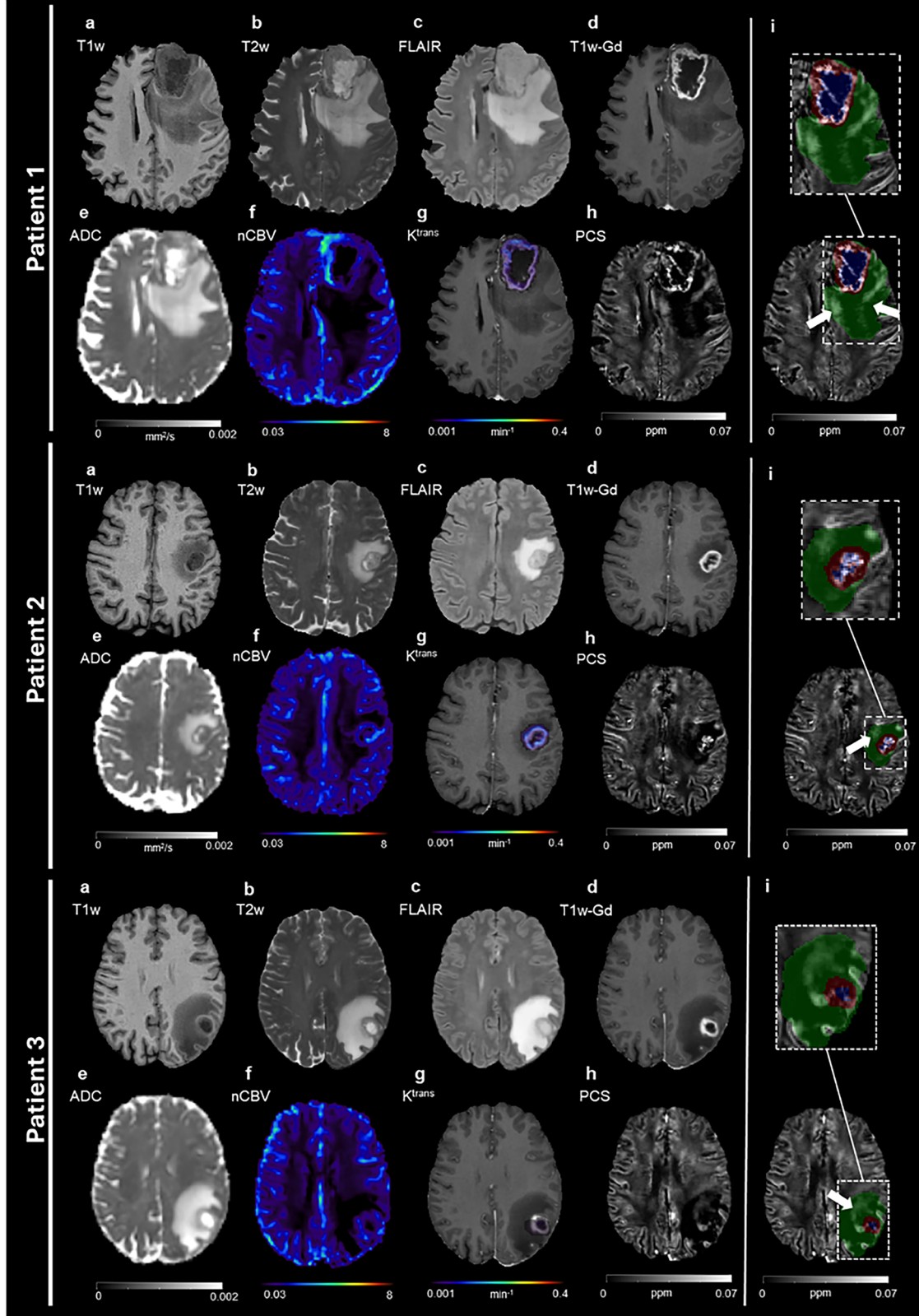

**Fig. 1 | PCS hyperintensity not visible on conventional MRI.** Axial slices of three representative glioblastoma patients. **a** T1w; **b** T2w; **c** FLAIR; **d** T1w-Gd; **e** ADC; **f** nCBV; **g** K$^{trans}$ (color scale) overlapped to the T1w-Gd (gray-scale); **h** PCS; **i** PCS on top of which the color-coded lesion segmentation masks are visualized and a zoomed view (dashed rectangle) of the lesion area (green: edema; blue: necrosis; red: enhancing tissue). White arrows indicate the PCS hyperintensity in edema.

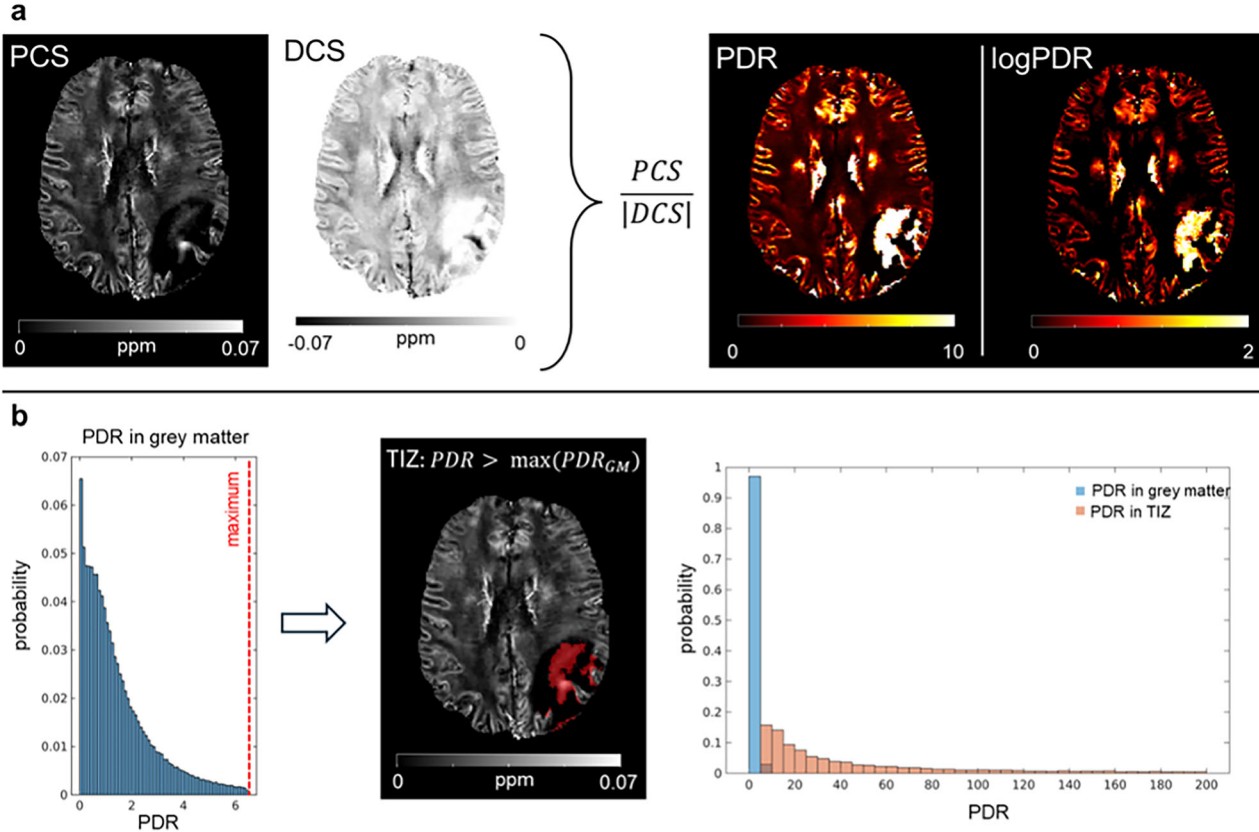

**Fig. 2 | PDR uniquely identifies the "TIZ" in edema. a** Example of PCS, DCS, PDR and logPDR maps. **b** The maximum of the PDR distribution of contralateral healthy gray matter (red dotted line on the histogram plot) is used as threshold to identify the "TIZ" (red area overlaid on the PCS in greyscale). The resulting distribution of PDR in "TIZ" compared to PDR in gray matter is reported on the right.

## PDR uniquely delineates the "TIZ"

Where PCS shows areas with distinctive hyperintensity in edema, identifying these zones automatically is challenging based on PCS alone, as these zones have PCS values similar to many other brain tissues, especially gray matter. Based on observations we had on the susceptibility maps derived from DECOMPOSE-QSM, we hypothesized that the ratio between the PCS and the DCS value would have been different between healthy tissues and the tumoral regions. Therefore, we computed the ratio between the PCS and the |DCS| (i.e., PDR) for each subject. The hyperintense-PCS zone in the edema exhibits unusually high PDR values. Further, these PDR values are dramatically higher than anywhere else within the brain parenchyma. In the tissue outside the tumoral lesion zone, the PDR in the apparently healthy GM exhibited higher values than apparently healthy WM for all subjects; therefore, we used the maximum values in GM as the threshold to identify automatically the "TIZ". Figure 2a shows an example of PCS and DCS maps from which the PDR was computed. By comparing the distributions of the PDR values in the "TIZ" and in the contralateral apparently healthy GM (Fig. 2b, histograms on the right), we observed that their range of values was vastly different. An example of "TIZ" is overlaid on the PCS for a representative patient (Fig. 2b, red area on the PCS). The PDR thresholds used for segmenting the "TIZ" can be summarized across patients as $9 \pm 3$ (mean ± standard deviation). The medians of the PDR in the "TIZ" across patients vary within the [17.29, 696.53] range, whereas the medians of the contralateral GM PDR are within the [0.82, 1.18] range. Statistically significant differences ($p < 0.05$) were found between distributions of PDR in the "TIZ" and the contralateral healthy tissues. Significant differences ($p < 0.05$) were also found between PDR values of voxels belonging to the "TIZ" and those in the rest of edema. As an additional way to assess the reproducibility of the "TIZ," we computed

the overlap between "TIZ" derived with STAR-QSM and iLSQR dipole inversion algorithms. The average Dice-Sørensen coefficient across subjects is equal to 0.7, indicating a good overlap between the two "TIZ" areas.

## PDR of "TIZ" in edema is associated with prognostic factors

We next asked the question of whether the PDR of the "TIZ" has biological relevance. We decided to compare the PDR of the "TIZ" with age and MGMT methylation. Age is a well-known negative prognostic factor since elderly patients have multiple comorbidities and an increased risk of adverse effects from radiotherapy. On the other hand, a positive MGMT methylation status is a positive prognostic factor as a decrease in MGMT expression improves the effect of treatment with alkylating agents[2]. Age and MGMT methylation are biological, generally accurate and reliable data specific for each subject. To investigate the associations between PDR in the "TIZ" and these biological variables, we first applied a logarithmic transformation to the PDR (logPDR) because the range of the PDR values can exceed a few orders of magnitude (Fig. 2b, histograms on the right). Then, a set of statistical features for the logPDR was selected for the analysis. Median and mode were included as summary statistics appropriate for non-normally distributed data. In addition, among the potential histogram features, the choice of the 10th percentile was data-driven, due to the specific distribution of PDR values in the "TIZ," entailing the vast majority of the data toward lower values and exhibiting a long right tail. This observation is still present even after the logarithmic transformation of the PDR values. The 10th percentile clearly differentiates the PDR of the "TIZ" from the range of the PDR values of the contralateral healthy tissues; thus, we hypothesized the potential of this feature as a marker given its specificity to distributions of the tumoral tissues. As reported in Fig. 3a, a significant association ($p < 0.05$) was found between age and the median of

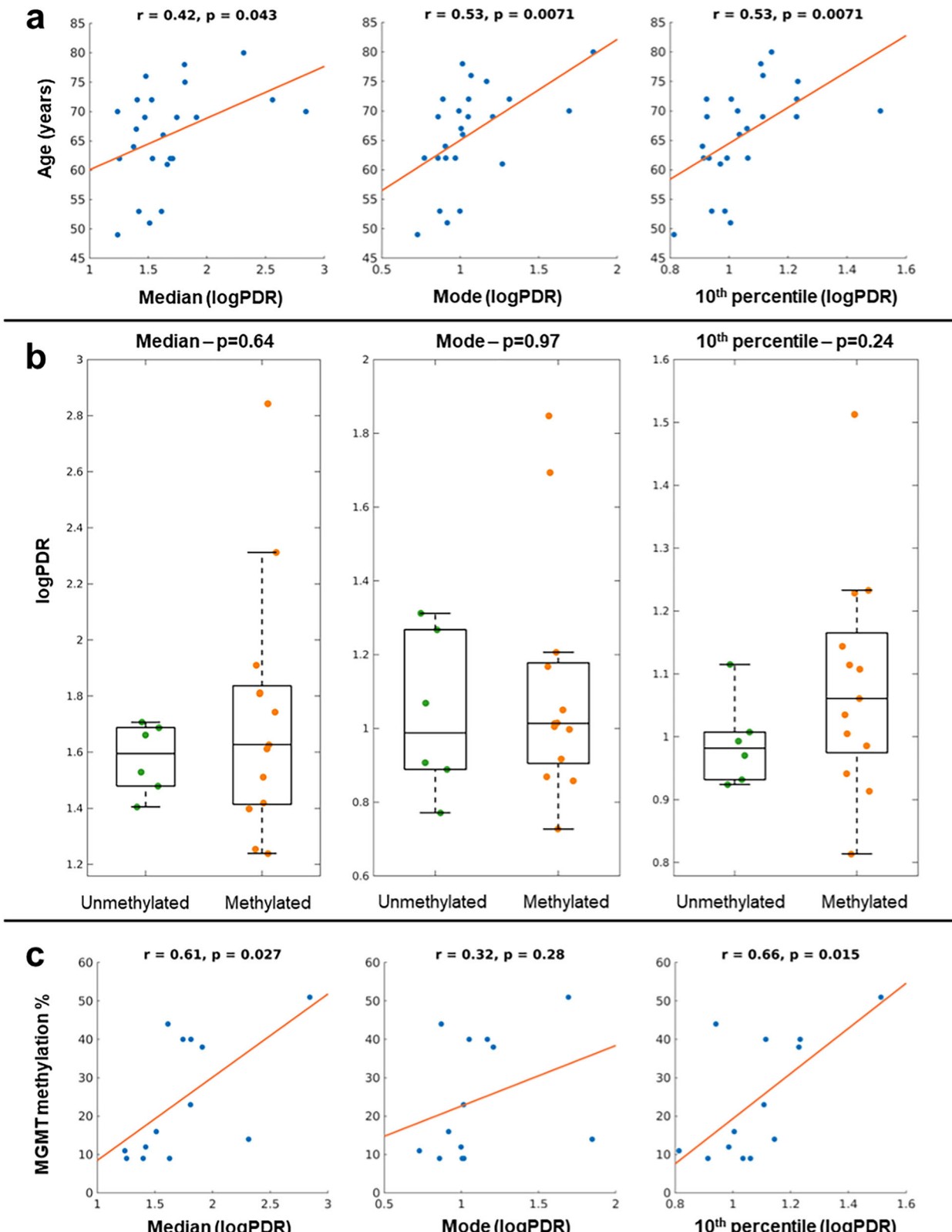

**Fig. 3 | Association of PDR in "TIZ" with prognostic factors. a** Associations between age and the features computed from the logPDR. **b** Group differences between patients with and without MGMT methylation. **c** Associations between the percentage of MGMT methylation and the features derived from the logPDR. In (**a**) and (**c**), r and p represent the correlation coefficient and the corresponding *p*-value, respectively. In (**b**), p represents the *p*-value of the statistical test.

the logPDR ($r = 0.42$), as well as the mode ($r = 0.53$) and the 10th percentile ($r = 0.53$). MGMT status, and consequently the percentage of methylation, were available for a subset of 19 patients, and 13 of them presented a positive methylation status. No statistically significant differences of

logPDR were observed between the methylated and unmethylated patient groups, for which boxplots are shown in Fig. 3b. Instead, significant correlations ($p < 0.05$) were found between the percentage of MGMT methylations and both the median ($r = 0.61$) and the 10th percentile

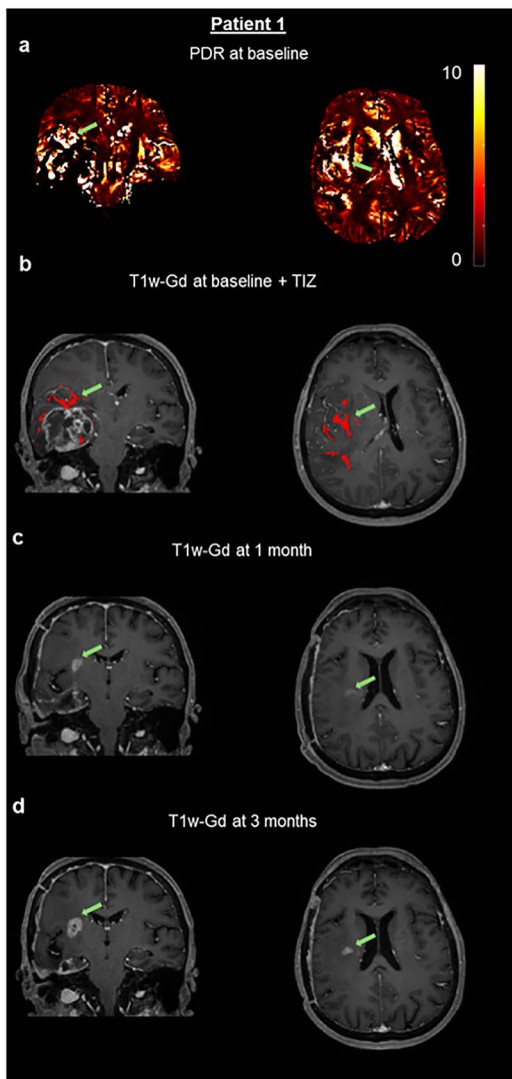

**Fig. 4 | PDR suggests tumor infiltration.** For one patient, PDR at baseline acquisition (**a**) from which the "TIZ" (red area) is segmented in edema, and it is overlaid on the T1w-Gd (**b**). One month after surgery, on the T1w-Gd (**c**), tumor enhancement appears at the same location indicated by the "TIZ" at baseline. Three months after surgery on the T1w-Gd (**d**), the tumor is growing at the same location. For another patient, PDR at three months after surgery suggests tumor infiltration (**e**) in an area of no contrast-enhancement on T1w-Gd at the location suggested by the "TIZ" (red area) (**f**). The corresponding location appears homogeneous on FLAIR (**g**). Four months after surgery on T1w-Gd, the tumor has invaded the location pointed by the "TIZ" (**h**). Green arrows indicate the location of tumor infiltration suggested by PDR. Images presented here are co-registered to the PDR image space of the corresponding patient.

($r = 0.66$) of the logPDR. Scatterplot of the data is presented in Fig. 3c. Interestingly, no significant correlation was found between age and the percentage of MGMT methylation. Given the well-known relationship between magnetic susceptibility and age during normal brain development and aging[49,51], we computed correlations between PCS values in the "TIZ" and age, but no significant associations were found, suggesting that the observed logPDR correlation with age is not caused by the normal aging process. In addition, no significant correlation was observed between the volume of the "TIZ" and that of edema, indicating that there is no influence of the volume on the segmentation of the "TIZ."

**Spatial distribution of PDR in edema suggests tumor infiltration**
To demonstrate the potential of PDR for detecting tumor infiltration, in Fig. 4, we report two patients as observational cases. For both patients, the evolution of the disease happened at a different location with respect to the one observed at the initial diagnosis. For one patient (Fig. 4a–d), PDR at baseline (Fig. 4a) shows specific areas in edema with very high values from which the "TIZ" is delineated. On the T1w-Gd of the same session, the location of the segmented "TIZ" (Fig. 4b,

red area) is above the contrast-enhancing tissue of the glioblastoma. One month later, at the location corresponding to one of those indicated by the "TIZ," contrast-enhancement appears on the T1w-Gd (Fig. 4c). Three months after surgery, the tumor progression is still ongoing at the same location (Fig. 4d). Note that there were other scattered "TIZ" voxels that did not lead to contrast-enhancement at one month nor at three months post-surgery. As the patient died shortly afterward, we do not have later time points to assess whether these "TIZ" voxels would lead to later enhancement. For another patient (Fig. 4e–h), PDR at three months shows a striking hyperintensity in areas of edema (Fig. 4e), and the corresponding "TIZ" is overlaid on the T1w-Gd (Fig. 4f, red area). T1w-Gd shows no contrast enhancement in the "TIZ," while FLAIR shows that the "TIZ" is within the edema tissue (Fig. 4g). One month later, contrast enhancement can be observed at the location indicated by the "TIZ" at the three-month MRI acquisition. The patient underwent a second surgery due to the suspicion of tumor localization. Glioblastoma was confirmed at pathology. Together, these observations suggest the potential of the PDR to track tumor infiltration through edema.

**Fig. 5 | Iron staining of tumor specimens confirms the biological origin of PDR.** Top row (**a**, **b**, **e**, **f**): presurgical in vivo MRI; Bottom row (**c**, **d**, **g**, **h**): histochemical iron staining and ex vivo MRI of tumor specimens. For Patient 1, the hyperintense FLAIR region around the tumor (**a**) exhibits elevated PDR (**b**) (arrows). Perls' staining for iron (**c**) of a tumor specimen shows high iron concentrations, where the corresponding ex vivo MRI scan presents high PDR (**d**) (arrows). For Patient 2, the boundaries of the tumor as delineated in the FLAIR image (**e**) present high PDR (**f**). Perls' staining for iron of the analyzed tumor specimen shows high iron concentrations in areas (**g**) corresponding to high PDR observed in the ex vivo MRI scan (**h**).

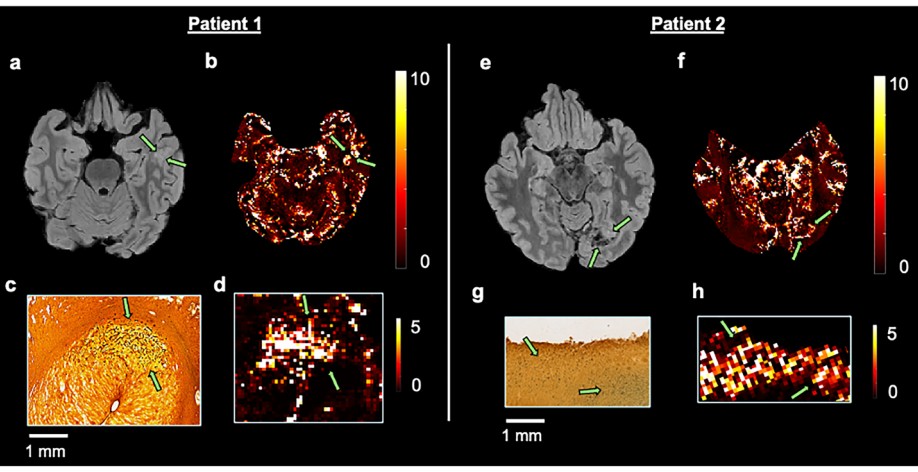

## Histochemistry and ex vivo MRI scans confirm iron as the main source for PDR

We confirmed iron as the main cause of the elevated PDR values for two independently acquired brain tumor patients (specifically, a neuroepithelial tumor and a ganglioglioma). For one patient, high PDR (Fig. 5b) can be observed in the area corresponding to the hyperintense FLAIR signal around the tumor (Fig. 5a). The ex vivo MRI scans of the surgically resected tumor specimen (Fig. 5d) show high PDR at the locations of high iron concentration according to the DAB-enhanced Perls' staining. For the second patient, at the boundaries of the tumor core visible on the FLAIR image (Fig. 5e), the corresponding PDR image shows high values (Fig. 5f). DAB-enhanced Perls' staining of the tumor specimen again revealed high iron concentration (Fig. 5g) at the location where high PDR was observed in the ex vivo scan (Fig. 5h). Together, these findings provide initial evidence of non-heme iron as the biological source of high PDR in the tumor.

## Discussion

Glioblastoma is managed with different treatment strategies, including surgical resection, radiotherapy and chemotherapy[2]. In clinical practice, several MRI modalities are employed for treatment planning and monitoring. Nevertheless, given the highly infiltrative nature of tumor cells, complete surgical resection of the tumor is not possible, and radiotherapy planning is not specific outside the tumor core. Previous imaging studies have investigated the heterogeneity of edema and the extent of the tumor outside the contrast-enhancing region[11,12,14–16,52], but there is still a lack of a sensitive and specific marker, which could also be applied at the single-patient level. Based on the hypothesis that the altered iron metabolism in the tumor could be detected with magnetic susceptibility imaging, we quantified the PCS by means of the DECOMPOSE-QSM model that separates paramagnetic and diamagnetic susceptibility contributions at the sub-voxel level. We propose a new method that specifically identifies areas of hyperintense paramagnetic susceptibility in the edema tissue of glioblastoma, and we introduce a new imaging marker of Tumor and Immune cell Infiltration Zone ("TIZ") in edema, based on the ratio of paramagnetic and diamagnetic components, i.e., PDR.

In this data-driven approach, we noted that the PDR of the PCS hyperintense area exhibited higher values compared to the contralateral healthy tissue. Therefore, we developed an automatic and reproducible procedure to segment the "TIZ" that is based on the definition of a threshold derived from PDR values in the apparently healthy contralateral tissues. GM segmentation included the cerebellum, which is known to be potentially affected by crossed cerebellar diaschisis[53], but there is no literature suggesting that it causes a change in QSM. Most importantly, the PDR in the "TIZ" is so much larger than GM (Fig. 2b) that PDR of the cerebellum would have a negligible effect on the choice of threshold. Nonetheless, additional

studies will be needed to fully assess the impact of segmentation of the cerebellum. Our methodology excludes the possibility that the "TIZ" is composed of healthy tissue only, given that healthy gray matter has significantly lower PDR. Of note, PDR exhibits a contrast that cannot be seen on conventional MRI modalities and that allows the delineation of the "TIZ" in the edema region. Further, the significant associations found between logPDR features and patient-specific biological variables suggest potential clinical utilities of PDR. Moreover, for the two observational cases we reported, PDR was able to predict the localization of tumor progression, which in fact overlapped with the location of the "TIZ." In their totality, our findings provide evidence that the PDR could be used as an imaging marker for tumor-associated cell infiltration and that PDR can identify spatial patterns of abnormalities in the edema tissue of glioblastoma.

Based on the results of the statistical analysis, the 10th percentile and the median of the logPDR in the "TIZ" were the most informative features for the associations with both age and the percentage of MGMT methylation. As hypothesized, the selected histogram features successfully captured the signature of the tumoral tissue underneath the "TIZ," which expresses magnetic susceptibility properties unique to the pathological tissue. The significant associations of logPDR features with these patient-specific factors further support its potential clinical relevance. MGMT methylation status is a positive prognostic factor as the decrease of MGMT expression improves the effect of treatment with alkylating agents[2], and patients with MGMT promoter methylation benefit more from combined radiochemotherapy[54]. Additionally, it has been previously reported that in a cohort of IDH-wt glioblastoma patients, MGMT methylation status presented intra-tumoral homogeneity[55]. Our data did not show significant logPDR differences between the unmethylated and methylated patient groups. The inability of logPDR features to differentiate between methylation status could be explained by the small and unbalanced sample sizes of the two groups. Instead, we observed a positive association between logPDR and the percentage of MGMT methylation. While not commonly used in clinical practice, the percentage of MGMT methylation has been proposed as a prognostic factor as well[56,57]. Although there is a lack of reported studies investigating the MGMT methylation relationship with immune-infiltrated cells, one study showed that tumors with a high MGMT mRNA expression (negatively correlated with MGMT methylation) exhibited gene signatures associated with various immune cells[58]. On the other hand, age is a well-known negative prognostic factor, and it has been shown that in the brain, immunosuppression increases during advanced age[59], and therefore, a higher logPDR feature value at older ages could be explained by an accumulation of TAMs. This agrees with the fact that a higher amount of immune infiltrate is associated with shorter survival[60,61] and that age is a negative prognostic factor. Though it is paramount to keep in mind that these observations reported in literature do not account for the complex

lesion heterogeneity, both in terms of the composition and spatial distribution of the tumor sample analyzed. Additionally, when interpreting the results, we must take into account that the number of patients included in the analysis of the MGMT methylation status and its percentage is lower than the one of the entire cohort, since for some of the patients the data were not available. In fact, no correlation was found between age and the MGMT methylation percentage for that subset of patients, indicating that the two variables are independent. Finally, the lack of a significant correlation between age and the PCS values in the "TIZ" suggests that susceptibility values in the "TIZ" are not driven by age, as happens for susceptibility of healthy tissues[49], which supports the pathological nature of the "TIZ" and that the abnormally high PDR in "TIZ" is caused by a short-term process most likely related to the disease itself.

PDR is defined as the ratio between paramagnetic susceptibility and diamagnetic susceptibility components of a voxel. The main biological source of paramagnetic susceptibility in the brain is iron, which can be found as non-heme- or heme-iron. Additionally, iron can be stored in ferritin within cells. Since both tumor and immune cells can be rich in iron content, the accumulation of iron makes the local magnetic susceptibility of tissues more paramagnetic, thus increasing PCS. An increased PCS is consistent with the higher PDR observed in the "TIZ". This was confirmed by histochemical analysis and ex vivo MRI acquisition of two tumor specimens, where elevated PDR was found in iron-rich areas, as highlighted by the stainings. On the other hand, the main source of diamagnetic susceptibility in the brain is myelin. Regions without myelinated neurons but with tumor and immune cells would thus have low DCS, thus high PDR due to the inverse relationship. Of note, the "TIZ" is the only area exhibiting PDR values higher than the threshold within the brain parenchyma, further suggesting its hypothesized pathological origin.

The main limitation of our study is the relatively small sample size. Although our hypothesis and findings are fully supported by the available data, the results must be validated in a larger and independent cohort of patients. The relatively small sample size has prevented us from fully assessing the sensitivity and specificity of PDR for predicting tumor progression. Nevertheless, there are two observations supporting the generalizability of our method. First, we observed the hyperintensity of the PCS and the PDR in edema consistently across all 24 subjects. Second, the PDR values of the "TIZ" are vastly different with respect to those of the contralateral healthy gray matter. If we perform a paired comparison between the PDR of the "TIZ" and that of the contralateral gray matter of the same subject, it's guaranteed that the "TIZ" has a larger PDR by how the "TIZ" is defined. On the other hand, if we were to perform an unpaired group comparison, we can consider the distribution of the median values of the PDR in the two regions and their standard deviations. It is possible to estimate the sample size required to detect significant differences between two groups with unequal variances. The average median for the "TIZ" across subjects is equal to 86.29, whereas for the contralateral healthy gray matter, it is equal to 1.04. The standard deviations are 149.84 and 0.08, respectively. Based on these values, at 5% confidence level and 80% of statistical power, the minimal sample size to detect the unpaired group difference is equal to 27. Of course, the estimated statistics are affected by the current sample size, and further studies with a larger sample size are indeed warranted. Moreover, further studies are needed to understand the biological processes underlying the hyperintense PCS and the PDR in the "TIZ". These studies ideally will identify the contribution from different cell types in detail.

To conclude, in the present work, we proposed and developed a new method for the identification of abnormal patterns in the edema region of glioblastoma that are most likely caused by cell infiltration. We developed a new tumor imaging marker called PDR based on the altered iron metabolism of the tumor. We found that PDR is biologically and pathologically relevant and can visually identify potential tumor and immune cell infiltration zone in the edema tissue. PDR can be readily generated from standard clinical MRI sequences. Our findings could provide guidance for both surgical and radiotherapy planning and treatment monitoring.

## Data availability

Data are available upon reasonable request from the corresponding author with an approved data agreement.

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

## Acknowledgements

G. Debiasi was supported by an AIRC Short-term Fellowship visit to UC Berkeley. C. Liu was supported by the US National Institute of Health through grants R01MH127104 and R01AG070826.

## Author contributions

A. Bertoldo contributed to conceptualization and data collection and supervision at the University of Padova. C. Liu contributed to the conceptualization and design of analysis approaches based on DECOMPOSE QSM, reconstruction codes and supervision at UC Berkeley. G. Debiasi and C. Liu contributed to data analysis, method for hyperintensity delineation and results interpretation at UC Berkeley. G. Debiasi contributed to data storage and organization at the University of Padova. G. Librizzi and R. Manara contributed to QSM sequence implementation, data acquisition, collection and automatic tumor mask check. G. Debiasi, V. Visani and M. Castellaro contributed to automatic tumor mask preparation. M. Castellaro contributed to QSM sequence implementation. Z. Li and H. Wei performed ex vivo MRI and histochemical analysis of tumor specimens. G. Librizzi and R. Manara contributed to clinical data collection and results interpretation. G. Debiasi and C. Liu wrote the original draft. A. Bertoldo contributed to results interpretation, reviewing and editing the original draft. All authors edited, reviewed and approved the manuscript before submission.

## Competing interests

The authors declare no competing interests.
