## [Transparent Peer Review file · Communications Medicine]

Magnetic susceptibility properties of tumor-associated cells imaged by MRI reveals glioblastoma infiltration in the edema region

Corresponding Author: Professor Chunlei Liu

Version 0:

Reviewer comments:

Reviewer #1

(Remarks to the Author)

The authors of the study attempted to develop a novel radiological technique that enables the identification of invaded tissues around the contrast-enhanced lesion for glioblastoma (GBM). They used Quantitative Susceptibility Mapping (QSM) based analysis as a backbone for diagnostic technology development. While the authors implemented challenging analyses, the following issues should be considered or clarified.

Methods

1. The assay system used to detect MGMTp methylation status should be described.
2. The authors state that GM and WM were automatically segmented by SPM12. The accuracy of segmentation should be critically evaluated, especially with brain images harboring tumors.
3. The authors also state that the tumor was automatically segmented. Its segmentation accuracy should be evaluated as well. Manual adjustment is usually necessary to ensure correct segmentation.
4. It is interesting to see that the authors attempted to eliminate tissues affected by Fe deriving from blood by using DCE and DSC perfusion images. However, the following statement is difficult to follow. Adding a graphical presentation is preferred. "initially, the TIZ in edema was determined by selecting the voxels whose PDR was above the maximum of the PDR between GM and WM PDR distributions."

Results

1. PCS maps are stated to have been calculated using two different methods. The results were visually presented using representative cases. A quantitative comparison using all cases should be much more reliable. One could perform a voxel-by-voxel comparison and calculate the difference between the two.
2. The actual results referring to "We further tested several other QSM algorithms" are not presented.
3. It is unclear why the authors chose to compare prognostic variables with PCS metrics deriving from the non-enhancing lesion. The biological characteristics of the tumor should be better represented by the tumor core.
4. The authors compared the effect of age and PCS metrics. If one would like to evaluate the correlation between age and any kind of radiological feature, a comparison within the normal-appearing brain would make more sense.
5. It is also difficult to understand why the authors compared MGMTp methylation status and PCS metrics. The MGMTp methylation magnitude is heterogeneous within the tumor, and conducting a linear correlation between it and a non-stereotactic image feature is undesired. Please refer to <https://academic.oup.com/neuro-oncology/article/21/5/616/5298599>
6. The name TIZ (Tumor and inflation zone) seems too bold based on a pathology observation of a single patient.

Reviewer #2

(Remarks to the Author)

The article introduces the PDR, a ratio between paramagnetic (PCS) and diamagnetic (DCS) susceptibility components, as a novel metric for uniquely delineating the hyperintense susceptibility area corresponding to a hypothesized Tumor and Immune cells Infiltration Zone (TIZ). The study employs various statistical methods to analyze the relationship between PDR and prognostic factors, such as age and MGMT methylation status.

This manuscript presents an innovative approach to glioblastoma imaging, leveraging PDR as a potential marker for tumor

and immune cell infiltration. The results are promising and highlight the potential clinical utility of PDR. However, the study has certain limitations, including insufficient elaboration on the biological underpinnings of PDR, a small dataset, and limited exploration of alternative metrics. Addressing these limitations may enhance the impact and clinical applicability of this work. Below are my specific comments:

1. One of my major concerns is the biological interpretability of PDR. While the pathological findings presented in this manuscript support PDR as a marker, they are primarily observational. The underlying biological basis for its specificity remains unclear. Could this specificity be inherently linked to the QSM technique itself? Providing a more detailed explanation of the physiological processes that connect the PDR value to tumor infiltration could significantly enhance the impact and robustness of this study.
2. The study includes only 24 patients, which limits its statistical power and raises concerns about the generalizability of the findings. It would be helpful to clarify whether this relatively small sample size is sufficient to confirm the sensitivity and specificity of PDR.
3. The PDR is defined as the ratio between PCS and DCS, which is a straightforward and intuitive metric. However, it may be worth exploring whether other mathematical formulations—such as combinations that account for the positive correlation of PCS and the negative correlation of DCS—might provide better specificity.

Reviewer #3

(Remarks to the Author)

This paper presents results from a preliminary study of 24 patients, newly diagnosed with glioblastoma, to explore the potential utility of quantifying the relative contributions of paramagnetic and diamagnetic susceptibility. The goal is to identify areas within edematous regions that may contain particularly high concentrations of invasive tumor cells. The method relies on iron metabolism, which is altered in tumorous cells compared to normal tissue.

The authors compute the Paramagnetic Component Susceptibility (PCS) from the patients' MR scans and demonstrate its hyperintensity in edematous regions. They perform additional analyses to show that the hyperintensity is not due to blood from leaky or increased vasculature. They also compute the Diamagnetic Component Susceptibility (DCS). The main finding of the paper is that the ratio of PCS to DCS can accurately identify regions of infiltrating tumor and immune cells. The findings are bolstered by evidence in specific cases where areas of high PCS/DCS outside the T1C enhancing region at initial diagnosis correspond to areas of tumor recurrence a few months hence and by a pathology sample obtained using a neuro-navigation system for reference.

I thought that the statistical analyses were appropriate for the given sample size.

Although the patient sample is small, I found the authors' results compelling, and they certainly warrant further study with a bigger set of patients.

Version 1:

Reviewer comments:

Reviewer #1

(Remarks to the Author)

The authors have sufficiently revised the manuscript.

Reviewer #2

(Remarks to the Author)

Thank you for addressing all my concerns. I have no further comments.

We would like to thank the Editor and the Reviewers for their time and the valuable comments and suggestions they provided to us. Please find point-by-point responses to the Reviewers' comments in blue color below. Changes in the manuscript have been highlighted in yellow.

Reviewer #1 (Remarks to the Author):

The authors of the study attempted to develop a novel radiological technique that enables the identification of invaded tissues around the contrast-enhanced lesion for glioblastoma (GBM). They used Quantitative Susceptibility Mapping (QSM) based analysis as a backbone for diagnostic technology development. While the authors implemented challenging analyses, the following issues should be considered or clarified.

We appreciate that the Reviewer acknowledges the challenge associated with our study and we thank the Reviewer for the issues brought to our attention. We will clarify each point with a detailed explanation.

Methods

1. The assay system used to detect MGMTp methylation status should be described.

We thank the Reviewer for the comment. The assay system used to detect MGMT methylation status has been added to the manuscript text (Methods section, *Biological relevance* paragraph). In particular, the following sentence was added "Methylation measurements were obtained by Pyrosequencing (PyroMark Q96 ID system, QIAGEN)."

2. The authors state that GM and WM were automatically segmented by SPM12. The accuracy of segmentation should be critically evaluated, especially with brain images harboring tumors.

We greatly appreciate the attention the Reviewer poses on this topic. Indeed, the presence of tumor may affect the quality of registration and segmentation. To account for this variability, we have excluded the lesion area during the estimation of the transformation. To further clarify, only the GM and WM segmentation belonging to the contralateral hemisphere were considered and each

segmentation was visually inspected and manually edited if clear mismatch was observed. These details are included in the *Structural images pre-processing* paragraph of the Methods section.

3. The authors also state that the tumor was automatically segmented. Its segmentation accuracy should be evaluated as well. Manual adjustment is usually necessary to ensure correct segmentation.

We thank the Reviewer for highlighting the importance of manual corrections in cases of automatic segmentations. We agree with the Reviewer on the potential inaccuracies derived from automatic delineation and for this reason every tumor mask was visually inspected and manually edited by neuroradiologists with experience in the glioma field, as indicated in the *Tumor segmentation* paragraph of the Methods section.

4. It is interesting to see that the authors attempted to eliminate tissues affected by Fe deriving from blood by using DCE and DSC perfusion images. However, the following statement is difficult to follow. Adding a graphical presentation is preferred. “initially, the TIZ in edema was determined by selecting the voxels whose PDR was above the maximum of the PDR between GM and WM PDR distributions.”

We thank the Reviewer for appreciating our approach. The confusing statement was revised as:
" Because the GM PDR is larger than the WM PDR, the maximum PDR of the remaining GM PDR distributions was used as the threshold to determine the “TIZ” (Fig. 2b). Specifically, for each voxel in the edema, if its PDR was above the threshold, it was assigned to the “TIZ” region; otherwise, it was not assigned to the “TIZ”.”

The revision is included in the *PDR computation and “TIZ” delineation* section. The procedure is illustrated in Fig. 2b.

Results

1. PCS maps are stated to have been calculated using two different methods. The results were visually presented using representative cases. A quantitative comparison using all cases should be much more reliable. One could perform a voxel-by-voxel comparison and calculate the difference between the two.

We thank the Reviewer for the suggestion. It is known that different QSM algorithms result in different quantitative results (see e.g. QSM Consensus Organization Committee, et al. "Recommended implementation of quantitative susceptibility mapping for clinical research in the brain: a consensus of the ISMRM electro-magnetic tissue properties study group." *Magnetic resonance in medicine* 91.5 (2024): 1834-1862. <https://doi.org/10.1002/mrm.30006>). For STAR-QSM and iLSQR algorithms this difference has already been evaluated, and it is less than 2% (Wei, Hongjiang, et al. "Streaking artifact reduction for quantitative susceptibility mapping of sources with large dynamic range." *NMR in Biomedicine* 28.10 (2015): 1294-1303. <https://doi.org/10.1002/nbm.3383>). Our aim here was to verify that the PCS hyperintensity we observed was not due to reconstruction artifact and ultimately the "TIZ" can be reproduced. Therefore, to address the Reviewer's comment, we carried out a quantitative evaluation by computing the Dice-Sørensen coefficient between the "TIZ" defined by STAR-QSM and iLSQR. Across subjects the average Dice-Sørensen coefficient is 0.7 which indicates a good overlap between the two "TIZ" areas, given that absolute magnetic susceptibility values are expected to be different. We have added this information to the Results and Methods sections.

2. The actual results referring to "We further tested several other QSM algorithms" are not presented.

We thank the Reviewer for bringing this point to our attention. Following the previous comment, our aim here was to confirm that the PCS hyperintensity in edema was not caused by reconstruction artifacts. We have added additional representative slices to Supplementary Figure S2 that show other QSM methods we tested for data reconstruction. It can be noted that absolute QSM values differ between the four methods, but this is in agreement with previous literature studies (see e.g. QSM Consensus Organization Committee, et al. "Recommended implementation of quantitative susceptibility mapping for clinical research in the brain: a consensus of the ISMRM electro-magnetic tissue properties study group." *Magnetic resonance in medicine* 91.5 (2024): 1834-1862. <https://doi.org/10.1002/mrm.30006>). Further, we have computed the Dice-Sørensen coefficients between the "TIZ" defined by two methods STAR-QSM and iLSQR. A thorough assessment of all existing QSM reconstruction methods is beyond the scope of the current study.

3. It is unclear why the authors chose to compare prognostic variables with PCS metrics deriving from the non-enhancing lesion. The biological characteristics of the tumor should be better represented by the tumor core.

We thank the Reviewer for the comment. We agree that the biological characteristics of the tumor should be generally better represented by the tumor core. To clarify, we are not trying to show that PCS or PDR metrics deriving from the edema provide a better characterization of the tumor itself. Rather, we are asking the question whether the PDR in the “TIZ” that we have identified has actual biological relevance. Therefore, we decided to investigate the relationships between prognostic variables and PDR related metrics deriving from the non-enhancing area, given that age and MGMT methylation status (and its percentage) are patient-specific and reliable biological measures. At the same time, they are clinically relevant prognostic factor and the relationship between them and the PDR in the “TIZ” is first evidence of the biological and clinical relevance of PDR as an imaging marker of the disease. Since our aim was to identify potential tumor-related activities in edema, our area of focus was outside the tumor core. We acknowledge that it would be interesting to compare results from both regions, but this goes beyond the purpose of the present study.

To avoid confusion, we have added the following sentences at the beginning of the Results section “PDR of “TIZ” in edema is associated with prognostic factors”:

“We next asked the question whether the PDR of the “TIZ” has biological relevance. We decided to compare the PDR of the “TIZ” with age and MGMT methylation.”

4. The authors compared the effect of age and PCS metrics. If one would like to evaluate the correlation between age and any kind of radiological feature, a comparison within the normal-appearing brain would make more sense.

We thank the Reviewer for the remark. We agree with the Reviewer. We and others have shown that QSM and PCS is correlated with age in normal-appearing brain, see for example,

Li, Wei, et al. "Differential developmental trajectories of magnetic susceptibility in human brain gray and white matter over the lifespan." Human brain mapping 35.6 (2014): 2698-2713.

Ahmed, Maruf, et al. "DECOMPOSE-QSM improves the tracking of iron-related neurodegenerative pathology." ISMRM 2023, abstract #2418.

Li, Zhenghao, et al. "APART-QSM: An improved sub-voxel quantitative susceptibility mapping for susceptibility source separation using an iterative data fitting method." Neuroimage 274 (2023): 120148.

Although it is known that paramagnetic susceptibility is positively associated with the healthy aging process, it is unknown if the PCS metrics derived from the "TIZ" is associated with age. Given that we didn't find any significant correlation in our analysis, we concluded that the tissue within the "TIZ" is indeed not driven by physiological processes associated with normal aging. On the contrary, this finding further supported the potential pathological nature of the phenomenon.

5. It is also difficult to understand why the authors compared MGMTp methylation status and PCS metrics. The MGMTp methylation magnitude is heterogeneous within the tumor, and conducting a linear correlation between it and a non-stereotactic image feature is undesired. Please refer to <https://academic.oup.com/neuro-oncology/article/21/5/616/5298599>

We thank the Reviewer for raising this consideration. We decided to include the MGMT methylation status and its percentage because they are patient-specific and are routinely used in clinical settings. The spatial localization of MGMT sampling was not recorded in routine clinical practice as conducted in our hospital. Furthermore, our patient cohort include only IDH-wt glioblastomas for which MGMT methylation based on pyrosequencing presents intra-tumoral homogeneity according to the suggested paper. The cited paper found heterogeneity only in IDH mutant patients. We have added this point to the Discussion section of the manuscript.

6. The name TIZ (Tumor and inflation zone) seems too bold based on a pathology observation of a single patient.

We appreciate the Reviewer opinion on our choice for the name of the hyperintense PCS area. We agree that the TIZ name is still a hypothesis, and it needs further validation. To drive home the hypothesized nature of TIZ, we modified the name to "TIZ" with the quotation mark to emphasize the preliminary nature.

Reviewer #2 (Remarks to the Author):

The article introduces the PDR, a ratio between paramagnetic (PCS) and diamagnetic (DCS) susceptibility components, as a novel metric for uniquely delineating the hyperintense susceptibility area corresponding to a hypothesized Tumor and Immune cells Infiltration Zone (TIZ). The study employs various statistical methods to analyze the relationship between PDR and prognostic factors, such as age and MGMT methylation status.

This manuscript presents an innovative approach to glioblastoma imaging, leveraging PDR as a potential marker for tumor and immune cell infiltration. The results are promising and highlight the potential clinical utility of PDR. However, the study has certain limitations, including insufficient elaboration on the biological underpinnings of PDR, a small dataset, and limited exploration of alternative metrics. Addressing these limitations may enhance the impact and clinical applicability of this work. Below are my specific comments:

We thank the Reviewer for the comment. We have addressed all the limitations and provided point-by-point replies in the following.

1. One of my major concerns is the biological interpretability of PDR. While the pathological findings presented in this manuscript support PDR as a marker, they are primarily observational. The underlying biological basis for its specificity remains unclear. Could this specificity be inherently linked to the QSM technique itself? Providing a more detailed explanation of the physiological processes that connect the PDR value to tumor infiltration could significantly enhance the impact and robustness of this study.

We thank the Reviewer for pointing out this concern. Indeed, further studies to fully assess the biological origin of PDR in a systematic way through tissue samples are needed. PDR is defined

as the ratio between paramagnetic susceptibility and diamagnetic susceptibility components of a voxel. The main biological source of paramagnetic susceptibility in the brain is iron, that can be found as non-heme- or heme-iron. Additionally, iron can be stored into ferritin within cells. Since both tumor and immune cells can be rich in iron content, the accumulation of iron makes the local magnetic susceptibility of tissues more paramagnetic, thus increasing PCS. An increased PCS is consistent with the higher PDR observed in the “TIZ”. On the other hand, the main source of diamagnetic susceptibility in the brain is myelin. Regions without myelinated neurons but with tumor and immune cells would thus have low DCS, thus high PDR due to the inverse relationship. Of note, the “TIZ” is the only area exhibiting PDR values higher than the threshold within the brain parenchyma, further suggesting its hypothesized pathological origin. We have added this explanation in the Discussion section of the manuscript.

2. The study includes only 24 patients, which limits its statistical power and raises concerns about the generalizability of the findings. It would be helpful to clarify whether this relatively small sample size is sufficient to confirm the sensitivity and specificity of PDR.

We thank the Reviewer for the remark, and we fully agree on the need of further studies to validate our findings and assess the generalizability on a larger and independent dataset. Nevertheless, there are two observations supporting the generalizability of our method. First, we observed the hyperintensity of the PCS and the PDR in edema consistently across all 24 subjects. Second, the PDR values of the “TIZ” are vastly different with respect to those of the contralateral healthy grey matter. If we perform a paired comparison between the PDR of the "TIZ" and that of the contralateral grey matter of the same subject, it's guaranteed that the "TIZ" has a larger PDR by how the "TIZ" is defined. On the other hand, if we were to perform an unpaired group comparison, we can consider the distribution of the median values of the PDR in the two regions and their standard deviations. It is possible to estimate the sample size required to detect significant differences between two groups with unequal variances. The average median for the “TIZ” across subjects is equal to 86.29, whereas for the contralateral healthy grey matter it is equal to 1.04. The standard deviations are 0.08 and 149.84, respectively. Based on these values, at 5% confidence level and 80% of statistical power, the minimal sample size to detect the unpaired group difference is equal to 27. Of course, the estimated statistics are affected by the current sample size, and further

studies with a larger sample size are indeed warranted. This clarification has been added in the Discussion section.

3. The PDR is defined as the ratio between PCS and DCS, which is a straightforward and intuitive metric. However, it may be worth exploring whether other mathematical formulations—such as combinations that account for the positive correlation of PCS and the negative correlation of DCS—might provide better specificity.

We thank the Reviewer for the comment. We agree with the Reviewer that there might be many other mathematical formulations to test. Though, as stated by the Reviewer, by taking the ratio of the susceptibility components we are introducing a metric that is easy to interpret and to understand. At the same time, it doesn't require any complex solution based on its formulation. Ultimately, a proper evaluation of the specificity of PDR needs to be addressed by further studies, ideally involving tissue sampling to better characterize the biological sources of susceptibility in the "TIZ".

Reviewer #3 (Remarks to the Author):

This paper presents results from a preliminary study of 24 patients, newly diagnosed with glioblastoma, to explore the potential utility of quantifying the relative contributions of paramagnetic and diamagnetic susceptibility. The goal is to identify areas within edematous regions that may contain particularly high concentrations of invasive tumor cells. The method relies on iron metabolism, which is altered in tumorous cells compared to normal tissue.

The authors compute the Paramagnetic Component Susceptibility (PCS) from the patients' MR scans and demonstrate its hyperintensity in edematous regions. They perform additional analyses to show that the hyperintensity is not due to blood from leaky or increased vasculature. They also compute the Diamagnetic Component Susceptibility (DCS). The main finding of the paper is that the ratio of PCS to DCS can accurately identify regions of infiltrating tumor and immune cells. The findings are bolstered by evidence in specific cases where areas of high PCS/DCS outside the TIC enhancing region at initial diagnosis correspond to areas of tumor recurrence a few months hence and by a pathology sample obtained using a neuro-navigation system for reference.

I thought that the statistical analyses were appropriate for the given sample size.

Although the patient sample is small, I found the authors' results compelling, and they certainly warrant further study with a bigger set of patients.

We thank the Reviewer for the comments. We agree with the Reviewer on the need of further studies including a bigger cohort of patients to validate our current findings.

We thank the reviewers for their continuing efforts.

In this revision, we replaced the original Figure 5 because some co-authors (who contributed to the pathological data in Figure 5) have a conflict of commitment due to their affiliation with a separate study. They discovered that they were not able to contribute that pathological data to this paper, as a result, they withdrew their names from this study. We discussed the situation with handling Editor Dr. Valerie Salvatico. The editors requested us to identify “another way to show this result and support the conclusion that iron depositions are located in the area examined via imaging”.

To obtain additional tissue samples and histology, we established a new collaboration with a separate hospital, which has allowed us to acquire data from two tumor patients. The two patients are of glioma types (grade 1) not glioblastoma (grade 4). We believe that the data strengthen the support for "iron depositions are located in the area examined via imaging" as the editors have requested. To summarize,

- 1) We performed *in-vivo*, pre-surgery MRI with the same methods described in the original paper. We computed the proposed imaging metrics (i.e. PDR) and replicated the finding of elevated PDR values.
- 2) To strengthen the data further, we went a step further and collected *ex vivo* MRI of the tissue samples this time and computed the PDR metric at higher spatial resolution that was not feasible *in-vivo*. The *ex-vivo* data again replicated the elevated PDR values.
- 3) We performed iron staining that clearly demonstrates the presence of high levels of iron at the regions of elevated PDR values.

We replaced the original Figure 5 with these new data.